# Extracellular miRNAs as Predictive Biomarkers for Glypican-3-Derived Peptide Vaccine Therapy Response in Ovarian Clear Cell Carcinoma

**DOI:** 10.3390/cancers13030550

**Published:** 2021-02-01

**Authors:** Mayu Ukai, Akira Yokoi, Kosuke Yoshida, Shiro Suzuki, Kiyosumi Shibata, Fumitaka Kikkawa, Tetsuya Nakatsura, Hiroaki Kajiyama

**Affiliations:** 1Department of Obstetrics and Gynecology, Nagoya University Graduate School of Medicine, 65 Tsuruma-cho, Showa-ku, Nagoya-shi, Aichi 466-8550, Japan; mayuoo@med.nagoya-u.ac.jp (M.U.); yoshida.kousuke@med.nagoya-u.ac.jp (K.Y.); kikkawaf@med.nagoya-u.ac.jp (F.K.); kajiyama@med.nagoya-u.ac.jp (H.K.); 2Institute for Advanced Research, Nagoya University, Furo-cho, Chikusa-ku, Nagoya-shi, Aichi 464-8601, Japan; 3Department of Gynecologic Oncology, Aichi Cancer Center Hospital, 1-1 Kanokoden, Chikusa-ku, Nagoya-shi, Aichi 464-8681, Japan; s.suzuki@aichi-cc.jp; 4Department of Obstetrics and Gynecology, Fujita Health University Bantane Hospital, 3-6-10, Otobashi, Nakagawa-ku, Nagoya-shi, Aichi 454-8509, Japan; shiba@fujita-hu.ac.jp; 5Division of Cancer Immunotherapy, Exploratory Oncology Research and Clinical Trial Center, National Cancer Center, 6-5-1 Kashiwanoha, Kashiwa-shi, Chiba 277-8577, Japan; tnakatsu@east.ncc.go.jp

**Keywords:** immunotherapy, peptide vaccine, ovarian cancer, miRNA, biomarker, ovarian clear cell carcinoma

## Abstract

**Simple Summary:**

The prognosis of ovarian clear cell carcinoma was poor due to chemoresistance; therefore, immunotherapy, such as the glypican-3 (GPC3) peptide vaccine, gained attraction, as it prolonged survival. There were some super responders; however, response rates were limited because of the lack of predictive biomarkers for the efficacy of treatment. The purpose of this study was to explore circulating biomarkers using pre-treatment serum samples from a GPC3 peptide vaccine clinical trial in order to predict response to vaccine therapy. We identified serum miR-375-3p, miR-193a-5p, and miR-1228-5p as predictive biomarkers of response to GPC3 peptide vaccine therapy. These miRNAs were not overexpressed in tumor tissues, and functional annotation analysis suggested that these miRNAs were associated with interferon-related pathways. Therefore, these biomarkers may reflect the immune activity of patients and may have broader applications in all immune-related therapies, including tumor vaccines.

**Abstract:**

Ovarian clear cell carcinoma (OCCC) has been treated with surgery and chemotherapy; however, the prognosis remains poor because of chemoresistance. Therefore, immunotherapies are attracting attention, including the GPC3 peptide vaccine, which improves overall survival. However, the response rate is limited and there are no sufficient predictive biomarkers that can identify responders before treatment. Our purpose was to identify circulating serum miRNAs as predictive biomarkers for response to GPC3 peptide vaccine. Eighty-four patients in a phase II trial of a GPC3 peptide vaccine were enrolled and miRNA sequencing was performed on their serum samples. Candidate miRNAs were selected from a group of 14 patients for whom treatment was responsive and validated in an independent group of 10 patients for whom treatment was responsive. Three markedly upregulated miRNAs, miR-375-3p, miR-193a-5p, and miR-1228-5p, were identified, and the combination of those miRNAs demonstrated high value in the prediction of the response. The origin of these miRNAs was assessed by referring to OCCC tissue miRNA profiles, and they were not identified as cancer tissue-related miRNAs. Functional annotation analysis suggested that they were associated with interferon-related pathways. The miRNAs identified herein have great potential to allow the realization of liquid biopsy for predicting the immunotherapy response and precision medicine.

## 1. Introduction

Epithelial ovarian carcinoma (EOC) is the most common type of ovarian cancer, and ovarian clear cell carcinoma (OCCC) is the histologic subtype, representing 3–25% of EOC worldwide, with a relatively poor prognosis [1,2]. Cytoreductive surgery and chemotherapy have been recognized as standard therapies for the last several decades. However, patients with OCCC tend to have more chemotherapy-resistant disease than other subtypes, resulting in poor prognosis [3]. Recently, anti-vascular endothelial growth factor antibodies and poly (ADP-ribose) polymerase inhibitors have been approved as new treatments [4,5,6,7]. These treatments significantly prolonged progression-free survival in trials, although there was no definitive increase in overall survival [1,4,5,6,7]. Therefore, immunotherapies received much attention as alternative strategies against ovarian cancers, and related therapies have been developed, such as cancer vaccines against neo-antigens, immune checkpoint inhibitors, and adoptive T-cell therapies [1]. Glypican-3 (GPC3), a heparan sulfate proteoglycan attached to the cell surface via glycosylphosphatidylinositol anchors, is overexpressed in OCCC and is a promising target for immunotherapy [8]. We have reported prolonged survival without severe toxicity in a phase II study examining the therapeutic effect of a GPC3 peptide vaccine in patients with OCCC [9]. Although there were some super responders to this vaccine, the response rate remained limited. This was mainly because it is difficult to predict the response before treatment. Therefore, accurate predictive biomarkers are needed for the realization of precision medicine targeting GPC3.

MicroRNAs (miRNAs) are small non-coding RNAs comprising 20–22 nucleotides and play important biological roles in diverse pathways [10]. MiRNAs have numerous functions in the cancer microenvironment and in the regulation of the immune system [11,12,13,14]. In addition, miRNAs can be encapsulated by extracellular vesicles and mediate cell-to-cell communication in the local and distant microenvironment [15,16,17]. Furthermore, extracellular circulating miRNAs are highly stable in biofluids protected from RNase degradation and have been variedly reported as biomarkers for early detection of cancer and prediction of treatment response [15,17,18]. Remarkably, serum miRNA profiles have been previously reported as promising diagnostic biomarkers in ovarian cancer [18]. Moreover, one of the advantages of serum miRNA analysis is the easy procurement of the whole profile by sequencing a small number of samples. Therefore, circulating miRNAs are potential diagnostic and predictive biomarkers for EOC [17,18].

In this study, we explore biomarkers accessible by less-invasive liquid biopsy for predicting the response to immune vaccine therapy using archival pre-treatment serum samples in a GPC3 peptide vaccine therapy clinical trial. In addition, we address the origin of candidate miRNAs and predict their functional pathways. The identified miRNAs have great potential to enable liquid biopsy for predicting the immunotherapy response.

## 2. Results

### 2.1. Study Design

In a phase II clinical trial of GPC3 peptide vaccine therapy between 2010 and 2016 at Nagoya University Hospital, 84 patients who received GPC3 peptide vaccine monotherapy after cytoreductive surgery plus adjuvant chemotherapy were enrolled in this study. To evaluate the usefulness of serum miRNAs as predictive biomarkers of therapeutic efficacy, 84 patients were split into two studies: study 1, remission group (*n* = 52), who experienced no recurrence between standard treatment and vaccination and study 2, progression group (*n* = 32), who have already relapsed before vaccination (Figure 1). Then, after eliminating patients according to the exclusion criteria, predictive biomarkers of the treatment response were investigated in the remission (*n* = 14) and progression (*n* = 10) groups by dividing them into responsive and unresponsive subgroups. The responsiveness was usually measured by computed tomography examinations and defined as follows: In study 1, patients with recurrence during vaccination were defined as unresponsive, whereas patients without recurrence during vaccination were defined as responsive; In study 2, super responders with tumor shrinkage were defined as responsive and patients who relapsed within 3 months were defined as unresponsive.

In study 1, candidate miRNAs were detected in the remission group, whereas in study 2, validation was performed in the progression group. In addition, we analyzed the expression of the miRNAs in OCCC tissues and evaluated whether miRNAs were dysregulated in the presence of tumors in study 3.

### 2.2. Study 1: Identification of Circulating miRNAs that Predict GPC3 Peptide Vaccine Efficacy

In study 1, 14 patients in remission were analyzed, and their characteristics are shown in Table 1. All patients in the responsive subgroup completed 10 cycles of the vaccination protocol, whereas the median number of vaccinations in the unresponsive subgroup was 6 (range, 3–8). The mean age in the unresponsive subgroup was significantly younger than that in the responsive subgroup (*p* = 0.017, Mann–Whitney U test); however, there were no significant differences in stage (*p* = 0.801, chi-squared test). MiRNA sequencing was performed on serum samples that were obtained between initial treatment and GPC3 vaccination (Figure 2A). Hierarchical clustering and heatmap analysis showed that miRNA profiles of the responsive subgroup, especially those of R1, R3, and R7, tended to be distinct from those in the unresponsive subgroup (Figure 2B). Moreover, in the principal component analysis (PCA), miRNA profiles were clearly separated between the two subgroups (Figure 2C). MiRNAs with absolute Log_2_ fold change of >1 between the responsive and unresponsive subgroups in the remission group were identified as candidate miRNAs, which included 32 upregulated and 45 downregulated miRNAs in the responsive subgroup compared with the unresponsive subgroup (Figure 2D,E). The normalized read counts of candidate miRNAs are presented in Appendix A.

### 2.3. Study 2: Validation of Candidate miRNAs in the Patients with Progressive Disease

To validate the 77 miRNA candidates identified in study 1, we performed miRNA sequencing using the serum samples of patients with progressive disease. In the progression group, no significant differences were observed in the patient background characteristics such as age (*p* = 0.383, Mann–Whitney U test), stage (*p* = 0.302, chi-squared test), and surgical procedure (*p* = 0.175, chi-squared test) between the responsive and unresponsive subgroups (Table 2). Even after receiving standard treatment, they showed disease progression and participated in the GPC3 peptide vaccine trial (Figure 3A). In some cases, the vaccine showed remarkable therapeutic effect, and they were able to receive the vaccine ≥13 times (Figure 3B and Table 2). In study 2, heatmap analysis and PCA revealed that miRNA profiles of the responsive subgroup were apparently not different from those of the unresponsive subgroup, and thus, miRNA profiles were not clearly associated with treatment effects (Figure 3C,D). In study 2, miRNAs with absolute Log_2_ fold change of >1 were also identified as in study 1. As a result, 48 and 61 miRNAs were increased or decreased more than two-fold in the responsive subgroup compared with the unresponsive subgroup, respectively (Appendix A).

Next, miRNAs that were upregulated or downregulated in both studies 1 and 2 were detected using the Venn diagram (Figure 4A). Among the 32 upregulated miRNAs in the responsive subgroup in study 1, eight miRNAs were also upregulated in the responsive subgroup of study 2 (Figure 4A). Similarly, among the 45 downregulated miRNAs in the responsive subgroup in study 1, eight miRNAs were also downregulated in the responsive subgroup of study 2 (Figure 4A). As the number of samples in the responsive subgroup of the progression group used in the analysis was very small (two samples), a significant difference could not be determined. Therefore, miRNAs that clearly differed between the two groups were identified. Then, three miRNAs, miR-375-3p, miR-193a-5p, and miR-1228-5p, were found because of a clear division in expression in study 2, in which the minimum value in the responsive subgroup was higher than the maximum value in the unresponsive subgroup (Figure 4B). These three miRNAs were also significantly different in the study 1 remission subgroup (Figure 4C). Finally, the predictive value of the three candidate miRNA biomarkers was validated using Receiver operating characteristics (ROC) curve analysis (Figure 4D). Area under the curve was 0.88 for miR-375-3p, 0.98 for miR-193a-5p, 0.95 for miR-1228-5p, and 1 for the three miRNAs combined. Each of the three miRNAs demonstrated high accuracy as predictive biomarkers. Moreover, accuracy was further increased when the three miRNAs were combined.

### 2.4. Study 3: Assessing the Function of Candidate miRNAs

To assess the functions of the three candidate miRNAs, we investigated miRNA expression in OCCC tissues. Sequencing of miRNAs was performed using paired cancer and contralateral normal ovarian tissues from 10 stage I OCCC cases. Heatmap analysis and PCA indicated distinct miRNA profiles between cancerous and normal ovarian tissues (Figure 5A,B). Compared with those in normal tissues, 43 miRNAs were significantly upregulated in cancer tissues. However, the three candidate miRNAs were not upregulated in cancer tissues, suggesting that these miRNAs were not commonly upregulated in tumors in general (Figure 5C). This means that the upregulation of these miRNAs could be due to some other factors, rather than tumor-related changes. To determine the function of the miRNAs, functional annotation analysis was performed using the miRsystem, revealing that there were six dysregulated reactome pathways (Figure 5D). Two immune-related pathways, including interferon signaling (*p* = 0.031) and interferon gamma signaling (*p* = 0.043), were significantly dysregulated. General cancer-related pathways were not included in this analysis. Through this analysis, involvement of these miRNAs in immune function rather than tumor tissue itself was indicated. 

## 3. Discussion

In this study, we identified that circulating miR-375-3p, miR-193a-5p, and miR-1228-5p have potential as predictive biomarkers of response to GPC3 peptide vaccine therapy, and these three miRNAs are not broadly upregulated in cancer tissue. Functional annotation analysis suggested that these miRNAs are associated with interferon-related pathways. Therefore, these biomarkers may reflect immune activity of the patient and have broader applicability to all immune-related therapy, including immune vaccine therapy.

EOC remains a gynecologic malignancy with one of the worst prognoses because of therapeutic resistance. More than 80% of patients with EOC initially respond to treatment; however, up to 70% of patients experience relapses within 12–18 months [1,19,20]. Eventually, cancer cells acquire resistance to various anti-cancer agents. Therefore, new treatment strategies, including immunotherapy, are desirable. A lack of intraepithelial tumor-infiltrating lymphocytes in EOC has been shown to be significantly associated with poorer survival, suggesting that immune function is associated with ovarian cancer prognosis [21]. Recently, immunotherapies, including cancer vaccines, adoptive T-cell transfer, and immune checkpoint inhibitors, have attracted attention as potential treatments to improve overall survival [9,22,23]. These immunotherapies represent different approaches. Cancer vaccines stimulate cancer antigen-specific T-cells to provide an anti-cancer effect. Alternatively, adoptive T-cell therapies activate lymphocytes with high anti-cancer activity in vitro, after which the lymphocytes are returned to the patient. Immune checkpoint inhibitors allow T-cells to kill tumor cells by blocking immune checkpoint molecules, such as programmed cell death 1 (PD-1) and PD ligand 1 (PD-L1) [24]. However, clinical trials of immunotherapy for ovarian cancer have shown relatively low response rates. For example, response rates for GPC3 peptide vaccine and antibodies inhibiting PD-1 or PD-L1 were approximately 6% and 6–15%, respectively [9,22,25]. Moreover, response rates to adoptive T-cell therapy have varied from 0% to 71%; however, even in a successful trial (five of seven cases (71%) responding), the duration of response was only 3–5 months [25,26]. Although the response rates to immunotherapies are limited, these therapies are associated with a robust and prolonged therapeutic effect compared with conventional treatment in some patients. Therefore, pre-treatment predictive biomarkers stratifying super responders are needed to maximize the effect of immunotherapy.

According to previous reports, miRNAs are intimately involved in cancer biology phenotypes, including tumor angiogenesis, chemoresistance, and immunoregulation of T-cell activation through regulation of the tumor microenvironment [13,27,28]. Therefore, miRNAs have potential applications as biomarkers and therapies in immunotherapy [24]. In the present study, we identified three extracellular miRNAs, miR-375-3p, miR-193a-5p, and miR-1228-5p, which predict the efficacy of GPC3 peptide vaccine therapy. Regarding the known functions of those targets, miR-375 has been reported to be involved in the immune response and as a predictive biomarker for therapeutic efficacy in various cancers. In head and neck squamous cell carcinoma, miR-375 acted as a modulator to increase the cellular immune response to cancer by inhibiting PD-1/PD-L1 signaling via the Janus kinase 2/signal transducer and activator of transcription 1 pathway [29]. Low miR-375 expression was shown to be a poor prognostic factor and was associated with chemotherapy resistance in small-cell lung cancer as well as prostate cancer [30,31]. Moreover, miR-375 was highly associated with treatment sensitivity in hepatocellular carcinoma, colorectal cancer, and breast cancer [32,33,34,35,36]. These studies demonstrate that high miR-375 may be associated with favorable outcomes, which is consistent with the results of the current study. The miR-193 family has also been shown to be associated with immune function. In multiple sclerosis, miR-193a was strongly upregulated in CD4+ lymphocytes in response to CD3/CD28 stimulation, suggesting miRNA-dependent regulatory involvement in immune function [37]. Moreover, miR-193b was upregulated during polarization to M2a macrophages [38]. In malignant pleural mesothelioma cell lines, miR-193a-3p suppressed the expression of PD-L1 by targeting its 3′ untranslated region [39]. However, according to previous reports, miR-193a-3p and miR-193a-5p promoted drug resistance in bladder cancer as well as prostate cancer [31,40]. MiR-193a-5p was reported to be upregulated in EOC with resistance to neoadjuvant chemotherapy, and miR-193a-5p and miR-193b-5p were associated with poor prognosis [41]. Therefore, miR-193a-5p was associated with immune function and may have reflected a background of chemotherapy resistance [41]. MiR-1228-3p was reported to stably exist in body fluids, and its expression level was associated with overall survival in non-small-cell lung cancer [42]. However, the relationship between miR-1228 and immune function remains unclear. Other miRNAs could also be relevant to the efficacy of cancer vaccines. In peptide vaccine therapy for colorectal cancer, high levels of miR-125b-1 and miR-378a in tumor tissue and miR-6826 and miR-6875 in plasma have been reported as predictive biomarkers of poor outcome [43,44]. In dendritic cell-based immunotherapy for renal cell carcinoma, miR-1186, miR-98, miR-5097, and miR-1942 were upregulated and miR-708 was downregulated in tumor cells that evaded immunotherapy [45]. Therefore, the optimal combination of miRNAs may depend on the cancer type and treatment methods. These findings fit with the current findings indicating that the miRNAs identified herein may reflect immune activation and immunotherapy response in OCCC.

Several limitations of this study should be acknowledged. First, the number of patients in this study was limited. This is because the GPC3 peptide vaccine is not yet a standard treatment for ovarian cancer, and the study sample was obtained from limited and valuable samples in the phase II study of the GPC3 peptide vaccine. This phase II study was conducted in a unique setting because patients were treated with single agents of neo-antigen vaccine, not combination therapy [9]. Therefore, we hypothesized that we can determine the direct effect of vaccine therapy in a human physiological setting. We believe that our findings promise to identify and predict patients who receive benefits from vaccine-induced immunity. In the remission group (study 1; patients who did not have measurable cancer), serum miRNA profiles were considered to be similar to their normal condition, whereas in the progression group (study 2; patients with measurable tumors), serum miRNA profiles were considered to be affected by both cancer itself and immune cells against cancer. Therefore, circulating miRNAs were evaluated in two cohorts with totally different miRNA profiles. We believe that this method of analysis can identify biomarkers that can be used in varied environments. Due to the small scale of this study, further, rigorous validation is necessary to prove that these miRNAs are clinically useful; however, this study has great implications and can provide new concept for immune vaccine therapy. Second, we focused on serum miRNAs in this study; however, other molecules, such as proteins and DNA, can also be attractive biomarkers and should be evaluated. Third, the detailed function of these miRNAs in immune response and tumor shrinkage remains unclear. We have previously confirmed that peptide-specific cytotoxic T-cell (CTL) clones can recognize and kill GPC3-positive OCCC cell lines [8]. Furthermore, we have demonstrated vaccine-induced immune reactivity against the GPC3 peptide by ex vivo IFN-γ ELISPOT analysis in two cases of super responders [46]. Therefore, we suspect that these miRNAs are derived from immune cells and reflect the patient’s immune status. Further studies on the origin and function of these miRNAs in immunity are required.

In conclusion, this is the first study that evaluated the usefulness of circulating miRNAs as predictive biomarkers for a peptide vaccine in ovarian cancer. Upregulated expression of serum miR-375-3p, miR-193a-5p, and miR-1228-5p may predict a favorable response to GPC3 peptide vaccines by reflecting the patient’s immune status. These miRNAs may contribute to the optimization of precision medicine and improve immunotherapy outcomes. Although the detailed role of each miRNA in the immune response remains unclear, future large-scale studies should be conducted to confirm the usefulness of these miRNAs as predictive biomarkers for selecting patients likely to respond to vaccine treatment.

## 4. Materials and Methods

### 4.1. Patients and Sample

Patients with OCCC who were enrolled in a phase II trial of GPC3 peptide vaccine therapy between 2010 and 2016 at Nagoya University Hospital were included in the study. A detailed study protocol was described in a previous report [9].

In study 1 of patients in remission, 52 patients who received GPC3 monotherapy after initial surgery and adjuvant chemotherapy were enrolled. The GPC3 peptide vaccination schedule for the remission group included a single vaccination every 2 weeks for a total of 10 vaccinations. Patients with the following characteristics were excluded: (i) no recurrence after GPC3 vaccine (these patients may be cancer free regardless of the treatment), (ii) no serum sample, (iii) no adjuvant chemotherapy, and (iv) unknown prognosis. The remaining 14 patients were divided into two groups: a responsive group (no recurrence during GPC3 peptide vaccine treatment, *n* = 7) and an unresponsive group (recurred during GPC3 peptide vaccine treatment, *n* = 7). Vaccination was discontinued when progressive disease (PD) was diagnosed using computed tomography (CT) or FDG positron emission tomography (PET)-CT scans; therefore, the number of vaccinations in the responsive and unresponsive groups differed.

In study 2 of patients with recurrence or relapse (progressive group), 32 patients were enrolled. The GPC3 peptide vaccine schedule for this group included one vaccination every 2 weeks for a total of six vaccinations, followed by one vaccination every 6 weeks. The vaccination was continued as long as it remained responsive. Patients with the following characteristics were excluded: (i) no serum sample, (ii) GPC3 peptide vaccine administered <3 times because it often takes a few months to induce an immune response, and (iii) relapse after 4 months due to the clinically slow-growing tumor, and a delayed effect of the vaccine was difficult to evaluate. The responsive subgroup consisted of three patients who were able to receive GPC3 peptide vaccine >10 times and showed partial responses, and the unresponsive subgroup consisted of seven patients who discontinued the treatment after 3–5 times due to tumor progression or decreased performance status.

Tumor regression was assessed using CT or FDG PET-CT scans before and after vaccination. Tumor responses were evaluated according to the Response Evaluation Criteria in Solid Tumors (RECIST) guideline (v1.1) [47]. Some patients in the progression group could not even undergo CT or PET-CT scans after vaccination due to the clinical cancer progression, and those patients were classified as unresponsive.

To evaluate the origin of circulating miRNAs, study 3 was performed using archival formalin-fixed paraffin-embedded (FFPE) tissues of OCCC. We randomly included 10 patients with stage I OCCC and used their paired cancer and contralateral normal ovarian tissues.

All serum samples were collected from patients who were registered in the clinical trial of GPC3 peptide vaccine therapy. Furthermore, all serum samples were drawn on the day of the first vaccination or within a month before the day. The trial was approved and monitored by the Institutional Review Board at Nagoya University School of Medicine and registered with the University Hospital Medical Information Network Clinical Trials Registry (UMIN-CTR number: 000003696). Moreover, the present study was approved by the ethics committee of Nagoya University Hospital (approval Nos. 2017–0053), and we obtained written informed consent from all patients.

### 4.2. RNA Extraction and miRNA Sequencing

Total RNA was extracted from 400 µL of each serum sample using the miRNeasy Serum/Plasma Kit (Qiagen, Hilden, Germany) and from eight 5-µm thick sections of FFPE blocks using the miRNeasy FFPE Kit (Qiagen). Small RNA libraries were prepared using the NEBNext Multiplex Small RNA Library Prep Set for Illumina (New England Biolabs, Ipswich, MA, USA). Polymerase chain reaction (PCR) products were purified using the QIAquick PCR Purification Kit (Qiagen), and electrophoresis was performed using 6% Tris-borate-EDTA gels at 120 V for 60 min. DNA fragments corresponding to 140–160 base pairs were trimmed. The DNA concentration was measured using the Qubit dsDNA HS Assay Kit and a Qubit2.0 Fluorometer (Life Technologies, Carlsbad, CA, USA). Finally, single-end reads were performed on the Illumina MiSeq (Illumina, San Diego, CA, USA).

### 4.3. Data Analysis

The CLC Genomics Workbench version 9.5.3 program (Qiagen) was used for analysis. Adapter sequences were trimmed, and sequences between 15 and 35 branch-point sequence were counted. The trimmed data were mapped to the miRbase 21 database, allowing two mismatches. Cases with low total read counts were excluded from analysis (R6 in Study 1 and P3, P4, and P9 in Study 2). Normalization was performed using reads per million mapped reads, and miRNAs with expression <100 reads in all samples were excluded.

Further analysis was performed using RStudio (RStudio, Boston, MA) and R software (version 3.5.0). The normalized data were converted to base-10 logarithms and z-scores. Hierarchical clustering analysis was performed by calculating the Spearman correlation coefficient and using the “ward.D2” clustering method. To generate a heatmap, the heatmap.2 function of the gplots package (version 3.0.3) was used. PCA was performed using prcomp, and the plot3d functions of the rgl package (version 0.100.30) were determined in R software. To identify dysregulated miRNAs in Studies 1 and 2, absolute Log_2_ fold changes >1 defined candidate miRNAs.

### 4.4. In Silico Analysis

To perform functional annotation, we used miRSystem (http://mirsystem.cgm.ntu.edu.tw/). Reactome pathway enrichment analysis was performed, and *p*-values < 0.05 were set as the cutoff.

### 4.5. Statistical Analysis

To examine miRNA expression differences in Study 3, multivariate analysis using the Wald test in DESeq2 (version1.8.0) was performed using RStudio. Then, miRNAs with absolute Log_2_ fold changes >2.5 and adjusted *p*-values < 0.05 were extracted.

Other statistical analyses were performed using IBM SPSS Statistics version 25 (IBM, New York, NY, USA). ROC curve analysis was performed to evaluate the accuracy of candidate miRNA prediction of therapeutic effect. A Mann–Whitney U test was used to compare continuous variables, and chi-squared tests were used to compare categorical variables; *p*-values < 0.05 were used to indicate statistical significance.

## 5. Conclusions

Upregulated expression of circulating serum miR-375-3p, miR-193a-5p, and miR-1228-5p may predict the response to GPC3 peptide vaccine for OCCC patients. In addition, these three miRNAs were not derived from the cancer tissue itself, and functional annotation analysis suggested that these miRNAs were associated with interferon-related pathways. Therefore, these biomarkers may reflect the immune activity of the patient and have broader application to all immune-related therapies, including cancer vaccines.

To the best of our knowledge, this is the first study to evaluate circulating miRNAs in patients prior to GPC3 peptide vaccine therapy for OCCC in order to identify predictive biomarkers of immunotherapy efficacy. Despite the major limitation of a small number of patients, the results of this study may lead to precision medicine and contribute to improved treatment outcomes for immunotherapy. Further large-scale studies are needed to investigate the detailed role of each miRNA in the immune response and to confirm the usefulness of each miRNA as a predictive biomarker for therapeutic efficacy.

## Figures and Tables

**Figure 1 cancers-13-00550-f001:**
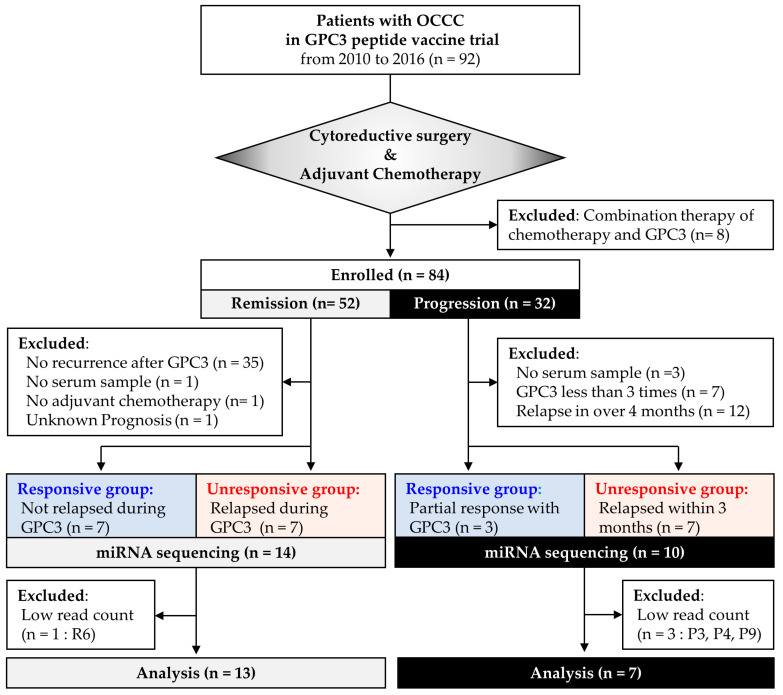
Patient flowchart. Patients with ovarian clear cell carcinoma enrolled in the glypican-3 trial were divided into remission and progression groups, and their archival serum samples were used for analysis. GPC3, glypican-3; R, remission; P, progression.

**Figure 2 cancers-13-00550-f002:**
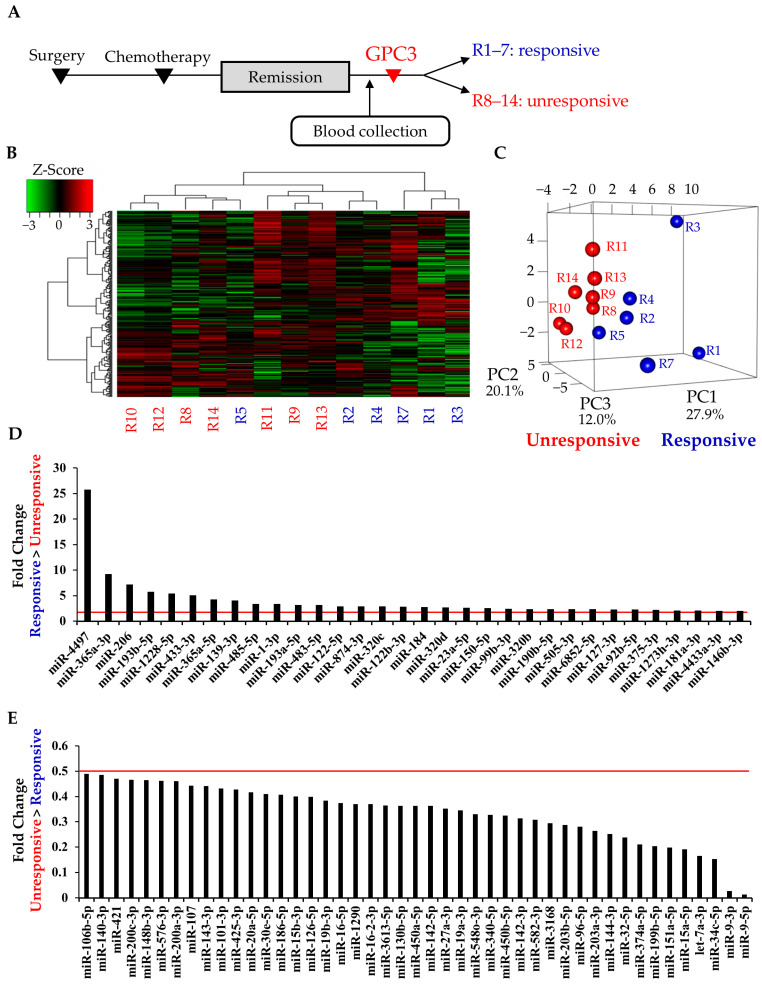
Comprehensive serum miRNA sequencing in the remission group. (**A**) Time line chart of the remission group; (**B**,**C**) heatmap and principal component analysis of serum miRNA profiles. Blue and red letters indicate responsive (R1–5 and R7) and unresponsive (R8–14) cases, respectively. R6 was excluded because of low read counts; (**D**,**E**) Up- and downregulated candidate miRNAs in the remission group. Red lines indicate absolute Log_2_ fold change of >1, fold change of >2 or <0.5; R, remission.

**Figure 3 cancers-13-00550-f003:**
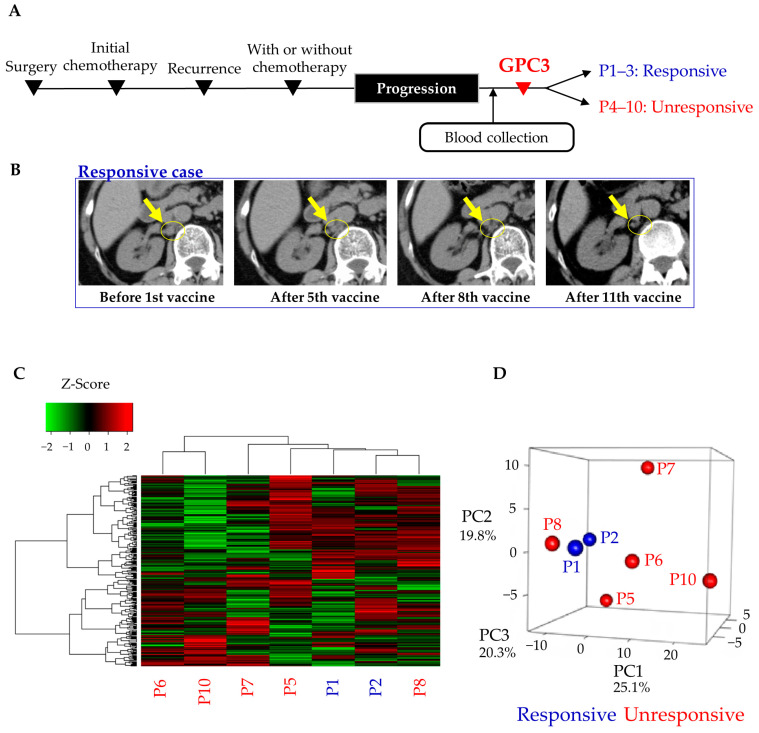
Comprehensive serum miRNA sequencing in the progression group. (**A**) Time line chart of the remission group; (**B**) representative images of a responsive case. A computed tomography scan showed that paraaortic lymph node metastases shrunk with GPC3 peptide vaccine treatment in the P2 case; (**C**,**D**) heatmap and principal component analysis of serum miRNA profiles. Blue and red letters indicate responsive (P1 and P2) and unresponsive (P5–8 and P10) cases, respectively. P3, P4, and P9 were excluded because of low read counts. P, progression.

**Figure 4 cancers-13-00550-f004:**
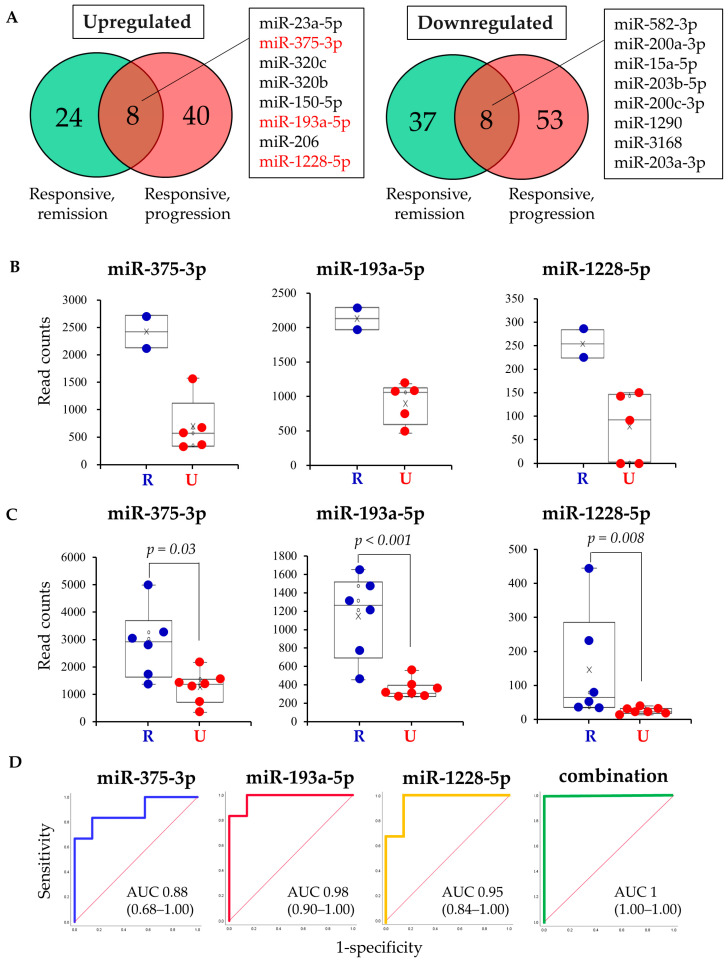
Extraction and evaluation of miRNAs as biomarkers. (**A**) Venn diagram showing commonly upregulated or downregulated miRNAs in Studies 1 and 2. Compared with each unresponsive subgroup, the number of miRNAs that were dysregulated in each responsive subgroup is shown. Responsive, remission: responsive subgroup in the remission group (Study 1); Responsive, progression: responsive subgroup in the progression group (Study 2); (**B**,**C**) box plots of expression of the three miRNAs in the progression and remission groups. The normalized read counts were compared using the Mann–Whitney U test. R: responsive, U: unresponsive; (**D**) ROC curves for detecting response to the GPC3 vaccine using each of the three miRNAs and combinations thereof.

**Figure 5 cancers-13-00550-f005:**
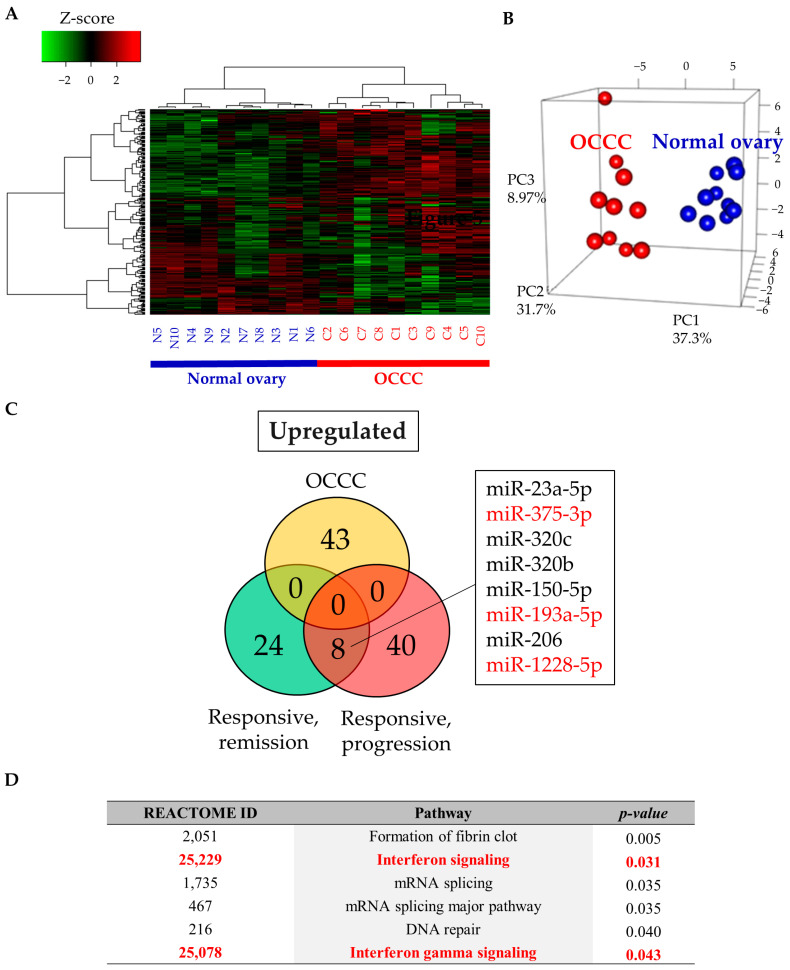
Comprehensive miRNA sequencing in FFPE tissues of OCCC and normal ovaries. (**A**,**B**) Heatmap and principal component analysis of tissue miRNA profiling. N: normal ovary, C: ovarian clear cell carcinoma; (**C**) Venn diagram showing upregulated miRNAs in OCCC FFPE tissue and serum samples (Studies 1 and 2). Responsive, remission: responsive subgroup in the remission group (Study 1), Responsive, progression: responsive subgroup in the progression group (Study 2); (**D**) functional annotation of the three miRNAs using miRSystem (http://mirsystem.cgm.ntu.edu.tw/).

**Table 1 cancers-13-00550-t001:** Characteristics of patients in remission.

Case	Age	Stage	Surgery	Chemotherapy	Number of Vaccination
Responsive subgroup					
R1	61	IIIc	ATH + BSO + OM + PEN + PAN	TC	10
R2	47	IIIc	ATH + BSO + OM + PEN + PAN	TC	10
R3	62	IIc(b)	ATH + BSO + OM + PEN + PAN	TC	10
R4	56	IIIc	RSO + OM	TC	10
R5	63	Ic(a)	BSO	TC	10
R6	56	Ic(b)	RSO + OM	TC	10
R7	59	IIIc	ATH + BSO + OM + PEN + PAN	TC	10
Unresponsive subgroup					
R8	49	IIIa	ATH + BSO + OM + PEN + PAN	TC	8
R9	51	IIc(1)	ATH + BSO + OM + PEN + PAN	TC	4
R10	40	IIIa	ATH + BSO + OM + PEN + PAN	TC	4
R11	54	IIIc	ATH + BSO + OM + PEN + PAN	TC, DC	3
R12	37	Ic(2)	ATH + BSO + OM + PEN + PAN	TC	8
R13	46	IIc(1)	ATH + BSO + OM + PEN + PAN	TC	6
R14	57	Ic(a)	ATH + BSO + OM + PEN + PAN	TC	7

ATH, abdominal total hysterectomy; BSO, bilateral salpingo-oophorectomy; RSO, right salpingo-oophorectomy; OM, omentectomy; PEN, pelvic lymph-node dissection; PAN, paraaortic lymph-node dissection; TC, paclitaxel and carboplatin; DC, docetaxel and carboplatin. The stage was indicated by International Federation of Gynecology and obstetrics (FIGO) 1988.

**Table 2 cancers-13-00550-t002:** Characteristics of patients in progression.

Case	Age	Stage	Surgery	Chemotherapy	Site of Recurrence	Number of Vaccination
Responsive subgroup						
P1	43	IIIc	ATH + BSO + OM + PEN + PAN	CPT-11 + CDDP	liver, retroperitoneal LN, peritoneum	13
P2	67	IIIc	ATH + BSO + OM + PEN + PAN	TC, CPT-11 + NDP, GEM + DTX	multiple LN	27
P3	65	IIc	ATH + BSO + OM + PEN + PAN	PLD + CBDCA	peritoneum	13
Unresponsive subgroup						
P4	50	IIIc	ATH + BSO + OM + PEN	TC, CPT-11, GEM	cancerous peritonitis	4
P5	51	IIIb	ATH + BSO + OM	TC, NGT, ETP	liver, multiple LN, peritoneum	3
P6	53	IIIb	ATH + BSO + OM	TCB, GEM	multiple LN peritoneum	3
P7	61	IIIc	ATH + BSO + OM	TCB, GEM	brain	3
P8	48	Ic	ATH + BSO + OM + PEN + PAN	TC, DC, PLD	liver, lung	4
P9	40	IVb	ATH + BSO + OM + PEN + PAN	TC, Radiation	bone	5
P10	63	IV	ATH + BSO + OM + PEN + PAN	TC, CPT-11 + CDDP, PLD, NGT, GEM, Radiation	multiple LN	3

ATH, abdominal total hysterectomy; BSO, bilateral salpingo-oophorectomy; OM, omentectomy; PEN, pelvic lymph-node dissection; PAN, paraaortic lymph-node dissection; LN, lymph node; CPT-11, irinotecan; CDDP, cisplatin; TC, paclitaxel and carboplatin; NDP, nedaplatin; GEM, gemcitabine; DTX, docetaxel; PLD, pegylated liposomal doxorubicin; CBDCA, carboplatin; NGT, Nogitecan Hydrochloride; ETP, etoposide; TCB, paclitaxel, carboplatin and bevacizumab; DC, docetaxel and carboplatin. The stage was indicated by the International Federation of Gynecology and obstetrics (FIGO) 1988.

## Data Availability

The data presented in this study are available on request from the corresponding author.

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
