# Peer review of "Extracellular miRNAs as Predictive Biomarkers for Glypican-3-Derived Peptide Vaccine Therapy Response in Ovarian Clear Cell Carcinoma"

_cancers, 2021, doi:10.3390/cancers13030550_

Round 1
Reviewer 1 Report
The authors have addressed my concerns. Their changes have significantly improved the manuscript and it's now clear that the work should be accepted by Cancers.
Reviewer 2 Report
This is a revised submission. The authors have been commendably thorough in addressing the concerns and questions of the reviewers, and the manuscript has been extensively revised and is much improved. I have no further concerns.
This manuscript is a resubmission of an earlier submission. The following is a list of the peer review reports and author responses from that submission.
Round 1
Reviewer 1 Report
The authors examine extracellular miRNAs as biomarkers to predict ovarian clear cell carcinoma patients who’ll respond to a vaccine derived from a Glypican-3 peptide. They identified 3 miRNAs that apparently predict response to the vaccine from sequencing of miRNAs from the serum of 24 subjects. They split their analysis between those who responded to surgery/chemotherapy (n=14) and those who did not (n=10). Data from the responding group was used to identify miRNAs, while data from the non-responders was used for validation. However, the rationale for this split is not provided and it’s not clear how comparison of people who did/didn’t relapse after surgery/chemo could be useful for identifying miRNA biomarkers that predict vaccine response. This is a weakness. Another weakness is the lack of explanation of the statistical processes used to establish that the power of their quite small groups was sufficient to identify statistically significant differences between subjects responsive to the vaccine and those who were unresponsive. Overall the data are not convincing that the identified miRNAs are useful predictors of vaccine response.
The following key issues should also be addressed.
- The rationale for focusing exclusively on miRNAs, and not other biomolecules like protein or DNA, requires explanation. In places, including the Simple Summary the authors propose that they explore “circulating biomarkers” but this is misleading because they only examine miRNAs. Why did they focus only on miRNAs - did they have an expectation based on the biology of this cancer that miRNA analysis would yield suitable biomarkers?
- The break-up of the patient cohort discussed in Fig 1 and the text at the bottom of page 2 is incomplete and a little confusing. The text should carefully explain what’s in the figure, but it doesn’t, and the terminology in the figure is unclear and the numbers don’t add.
Enrolment was of 84 patients treated with surgery and chemotherapy. These split into 52 “Remission” and 32 “Progression” presumably meaning that the 52 subjects didn’t experience a recurrence during the period between their surgery/chemotherapy and subsequent vaccine, while the 32 Progression cases did recur between surgery/chemo and when the vaccine was administered. Please clarify this.
Of the 52 Remission, 37 are excluded for one of three reasons, leaving 15 subjects, but the table shows that miRNA sequencing was only performed on 14 subjects – what happened to the 15th subject? Of the 14 remaining subjects, 7 were classified as Responsive to the vaccine and the other 7 as Unresponsive (but one of the responsives was excluded because of low miRNA sequencing count), but how responsiveness was measured was not explained – while this detail is likely in their trial paper, the authors should provide relevant information briefly in this manuscript. The group sizes used for this discovery phase are quite small (n=7 and n=6) – it would be helpful for the reader to see more detail about how statistical analyses were performed.
Of the 32 Progression, 22 are excluded for one of three reasons, leaving 10 subjects. The authors should clearly explain the rationale for excluding 12 subjects because they relapsed in over 4 months. This group of 12 is about the size of the Study 1 (n=14) and Study 2 (n=10) cohorts so their analysis could be expected to provide useful information. Similar to the Remission cases, of the 10 remaining subjects, 3 were classified as Responsive to the vaccine and the other 7 as Unresponsive, but how responsiveness was measured was not explained - the authors should provide that information.
- The rationale for splitting into two Studies isn’t clear. The authors need to explain how the comparison of vaccine “responsive” and “non-responsive” subjects who were in remission after surgery/chemotherapy is better than a pooled approach in which all “responsive” cases are compared with all non-responsive” cases irrespective of whether they relapsed between surgery/chemotherapy and the time the vaccine was administered.
- There are large differences in number of vaccinations received by patients and the reason for this is not explained. In both Table 1 and Table 2 the patients who received more vaccinations were in the “Responsive” groups. This should be explained.
- The authors propose that hierarchical clustering in Fig 2B shows that responsive R1, R3 and R7 are “remarkably distinct” from the unresponsive group – but this is certainly not apparent by eye. To my eye the most similar are R11 and R13 which are in the unresponsive group. I think that this qualitative description by the authors highlights a key weakness of the manuscript – the lack of quantitative statistical analysis. If the authors want to confirm that the subgroup of R1, R3 and R7 are “remarkably distinct” they must provide quantitative data, and to show that this sub-grouping is relevant the analysis needs to be supported by well described statistical analysis. The data in Fig 2D and 2E should be displayed as boxplots (as done in Fig 4B & C) with p values to indicate which differences are statistically significant between groups.
- Analysis of samples from the Progession group is also poor. At the start of section 2.3 the authors propose that analysis of this group is to “validate the 77 miRNA candidates identified in Study 1” - but that isn’t the case because they merely perform miRNA sequencing of patient samples as they did for Study 1. Their approach suggests that they should combine study 1 and study 2.
Also, the authors should explain what is meant by the statement “In the progression group, there were no significant differences between responsive and unresponsive subgroups (Table 2)”? What analyses did they perform to compare the subgroups? Is it surprising that there were no differences when there’s only 3 subjects in the Responsive group and 7 in the Unresponsive group? Similarly, the following statement also requires explanation “Heatmap analysis and PCA revealed that serum miRNA profiles of patients with progressive disease were not clearly associated with therapeutic effect (Figure 3C and 3D).” Without explanation in the text it’s not possible to understand how the data in those two panels show no clear association with therapeutic effect. In addition, the authors should provide a supplementary table listing the data for the identified 48 and 61 miRNAs that were increased or decreased.
The authors focus on 3 miRNAs (miR-375-3p, miR-193a-133 5p, miR-1228-5p) identified from Study 2 but there’s no discussion of how they relate to Study 1 – in fact the values shown in Fig 4C for the remission group don’t appear to the values displayed differently in the graph in Fig 2D. Also The graphs in Fig 4B lack p values. In Fig 4B and 4C what do “N” and “C” mean?
Other issues:
1. The wrong tense is used in the Simple Summary and the Abstract.
2. In the first sentence of the Introduction EOC is described as a lethal gynecologic malignancy. This implies that it is always lethal – which it is not. This sentence should be changed to avoid exaggeration.
Reviewer 2 Report
The study attempted to identify circulating biomarkers among patients participating in a GPC3 peptide vaccine clinical trial to predict response to the vaccine therapy. Three serum miRNAs were found to be predictive biomarkers of response to the vaccine.
This manuscript provides clear description of methods and presentation of results. Further elaboration is necessary.
- While miRNAs are known to be important. The authors, however, have not explained why they chose to evaluate miRNA profile as candidates of predictive markers.
- They have also not extrapolated on the biological basis of why these three miRNAs can act as predictive markers. Is it related to the downstream genes that they regulate?
Reviewer 3 Report
This is a well presented and interesting study on the predictive value of serum miRNA on response to glypican-3-derived peptide vaccination in patients with clear cell ovarian cancer. One of the key strengths is the number of patients enrolled in a vaccine trial for a less commonly seen pathologic type of ovarian cancer. As far as I can judge, serum samples drawn between initial treatment and GP3C vaccination, but the precise timing is not clear throughout the analysis. The authors identify 3 miRNAs, 375-3p, 193a-5p and 1228-5p, that may have predictive value of clinical response to GP3C vaccination. The data interpretation appears to be reasonable, and the discussion is well written with respect to related studies but does not provide a detailed critique of the results in this manuscript.
The overall conclusion is that the biomarker miRNAs are more likely associated with immune response pathways than tumor response to GP3C vaccination. This is probably correct, but it begs the question of whether or not these miRNAs correlated with immune response to GP3C vaccination, and whether immune responses were associated with clinical responses. This information is not presented, but is important for interpretation of the value of these miRNAs as valid biomarkers. The overall impression is that the results may have more bearing on immune response to GP3C vaccination rather than clinical response in OCCC patients, but this is hard to judge without information on vaccine immune correlates with clinical response.
I have some concerns with lack of clarity in the figure legends. For example, the graph labels E and N in Figure 4B and C are not defined. Also, I'm not clear on the definitions of responsive remission versus responsive progression in the Venn diagrams for Figure 4A and Figure 5C. The data in Fig 5C refer to miRNA in FFPE, but it's not obvious what these data represent with respect to responsive remission versus progression. The authors state that the miRNAs were not upregulated in cancer tissues, but they pick out eight miRNAs (including the three candidate miRNAs of interest) at the overlap between the response remission versus progression groups in the Venn diagram. This is confusing, and should be more fully explained.